# An accurate wearable hydration sensor: Real-world evaluation of practical use

Dmitry Rodin[1☯], Yair Shapiro[2☯], Albert Pinhasov[3☯], Anatoly Kreinin[3☯], Michael Kirby[3☯] *

1 Institute for Personalized and Translational Medicine, Ariel University, Ariel, Israel, 2 Department of Health Administration and Adelson School of Medicine, Ariel University, Ariel, Israel, 3 Department of Molecular Biology and Adelson School of Medicine, Ariel University, Ariel, Israel

☯ These authors contributed equally to this work.
* michael.kirby566@gmail.com

**Data Availability Statement:** Non-identifying human participant data is available in a public dataset archive. SpectroPhon DBM Subject Data. Published: 6 September 2021, Version 1, DOI: 10.

## Abstract

A wearable body hydration sensor employing photoplethysmographic and galvanic biosensors was field evaluated using 240 human participants with equal numbers of men and women volunteers. Monitoring of water mass loss due to perspiration was performed by medical balance measurements following one of two different treadmill physical exercise regimens over 90 minutes in 15-minute intervals with intervening 10-minute rest periods. Participants wore two different models of the dehydration body monitor device mated to commercially-available smartwatches (Samsung Gear S2 and Samsung Gear Fit2). Device output was recorded by Bluetooth wireless link to a standard smartphone in 20-second blocks. Comparison of the devices with the standard measurement method (change in body mass measured by medical balance) indicated very close agreement between changes in body water mass and device output (percent normalized mean root square error averaged approximately 2% for all participants). Bland-Altman analyses of method agreement indicated that <5% of participant values fell outside of the 95% confidence interval limits of agreement and all measured value differences were normally distributed around the line of equality. The results of this first-ever field trial of a practical, wearable hydration monitor suggests that this device will be a reliable tool to aid in geriatric hydration monitoring and physical training scenarios.

## Introduction

Adequate hydration is essential for good health and aids in support of all body systems. Low amounts of water hydration have a variety of negative health effects: mild dehydration manifests in headache, tiredness and thirst while severe cases may lead to fever, hypotension, rapid heart rate, increased respiration, cognitive impairment, and even unconsciousness [1–3]. States of dehydration drive thirst response through several sensory mechanisms including hypothalamic osmoreceptors, increases in gastric sodium ions and osmolality, and reduced blood pressure and volume stimulating antidiuretic compensatory increases in antidiuretic hormone, angiotensin, and renin secretion [4]. However, in thirst response in humans is more

17632/jt22782wjh.1 https://data.mendeley.com/
datasets/jt22782wjh/1.

**Funding:** The authors received no specific funding
for this work.

**Competing interests:** The authors have declared
that no competing interests exist.

complicated and is often mitigated by situational and conditioned behavioral factors such as
fluid taste and availability, linked association of drinking with meal times, and patterned drink-
ing habits [5], all of which can suppress perception of thirst. Furthermore, several studies have
demonstrated that conscious perception of thirst has a poor correlation with blood osmolality
[6]. Thirst response alone has been shown to be an inaccurate indication of hydration need [7,
8]. Therefore, there are a variety of situations, such as athletic training or status monitoring of
hospice patients, where a method of measuring hydration or water loss would be useful.

Several types of devices have been designed to monitor drinking frequency and amount in
elderly hospice patients who often will forget to drink [9]. These devices typically involve either
a smartcup that measures fluid volume removal or consists of a wearable inertial sensor that
detects specific wrist movements associated with drinking from a container [10, 11]. These are
inherently problematic as they indirectly measure drinking behavior and not actual hydration.
For people engaged in fitness or for professional athletes, the need for proper hydration moni-
toring is evident but until recently there was a lack of convenient devices to fulfill this need.
Efforts have been made to develop devices around materials technologies using microcapillary
sweat collection systems for volumetric estimates [12, 13] which are still in their development
stages and are prone to variance due to changing environmental use conditions. The portabil-
ity potential of fluid collection-based perspiration monitoring devices as wearables, whether
designed on microcapillary collectors or absorbent pads, is foiled by their inherent bulkiness
and power requirements.

Wearable electrochemical or optical sensors have the potential for repeated use and device
accuracy in producing a useable tool for real-time perspiration monitoring. SpectroPhon
LTD has developed a technology that allows measurement of very small amounts of solutes
contained in sweat using photoplethysmographic sensors covered with a special coating, the
outputs of which are deciphered by unique algorithms. These biosensors can be easily incor-
porated into most commercially-available smartwatches or smartbands for real-time mea-
surements synchronized to consumer smartphones with health monitoring applications.
The present work constitutes the first-ever independently-conducted field test of a wearable
hydration monitor commercial prototype with human volunteers. The main objective of cur-
rent study is to estimate the accuracy of SpectroPhon perspiration biosensors incorporated
in two smartwatches: a Samsung Gear S2 and a Samsung Gear Fit2. The secondary aim of the
study is to also evaluate the safety-in-use of SpectroPhon biosensors.

## Methods

### Tested device

The Dehydration Body Monitor (DBM) model SP-DBM (Firmware v1.5, SpectroPhon, Ltd.,
Rehovot, Israel) is a label-like, thin layer device affixed to the case back-glass of a smartwatch
with a modified and accessible CMOS interface (Fig 1A). The SpectroPhon devices use a pro-
prietary photoplethysmographic sensor (US20150260656A1, US20170027482A1; patents
pending) that changes optical characteristics in the presence of different metabolites in sweat
(water, lactic acid, pyruvic acid, carbonates, ketones, and monovalent ions such as sodium and
potassium). Differing concentrations of sweat metabolites affect the chemochromic character-
istics of the device and alter signal throughput, which is recorded, algorithmically transformed
in the smartwatch, and transmitted by Bluetooth to a synchronized smartphone. The DBM
specifically is attuned to detect various salts in secreted sweat, estimating sweat volume using
proprietary algorithms, and also employs a galvanic contact system to estimate whole body
skin surface area, which is used to estimate total body water loss. A data flow diagram is pre-
sented in Fig 1B.

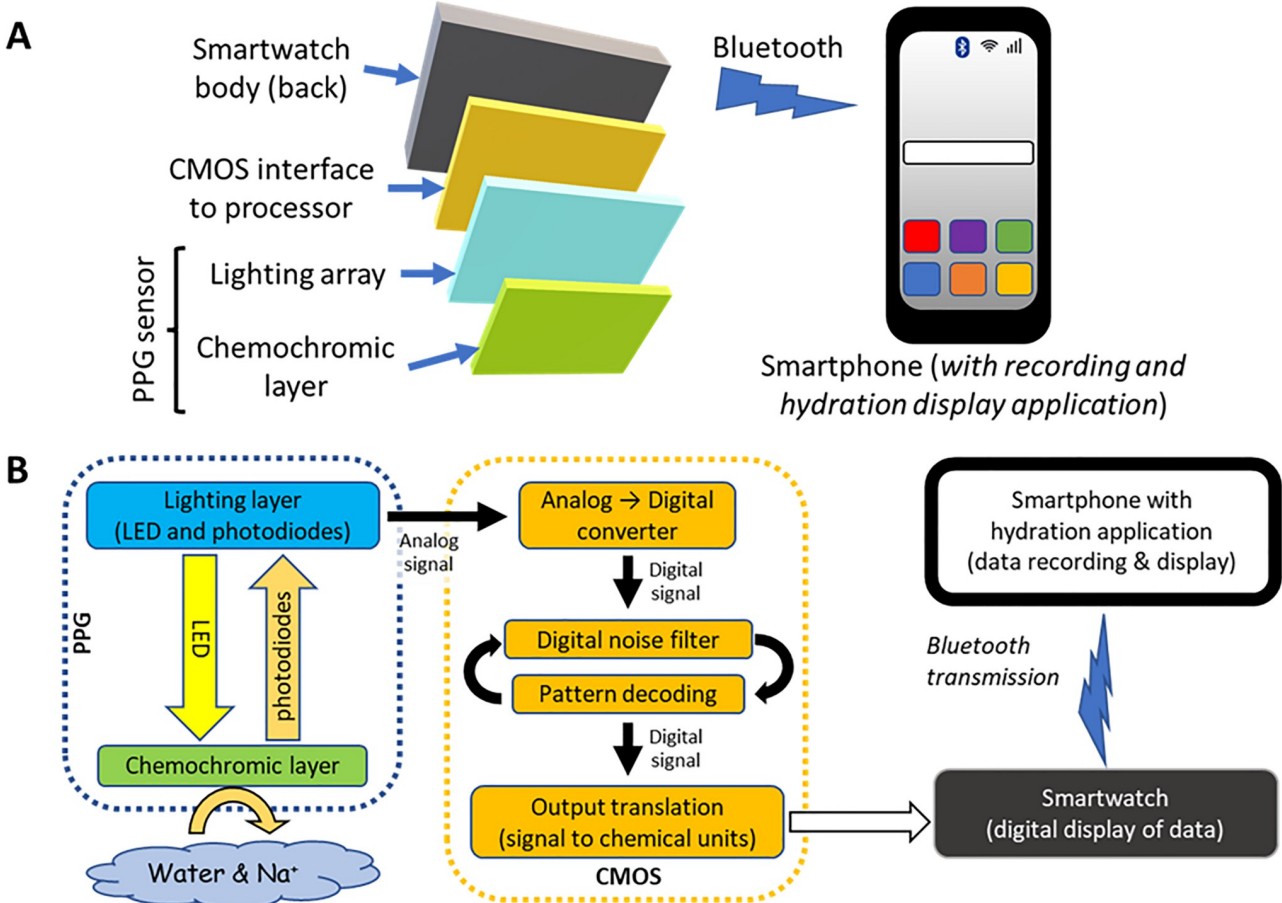

**Fig 1. Structural components and data flow of the SpectroPhon-DBM.** (A) Component arrangements of PPG sensor and CMOS interface to a smartwatch, Bluetooth-linked to a smartphone with data interpretation application. (B) Data flow diagram of components. Colors represent: green, chemochromic film; blue, lighting layer consisting of LED emitters and photodiode detectors; orange, components of the CMOS interface.

## Experimental design

Healthy adults (n = 240) of both sexes in different age groups were recruited and consented for the study by informed written consent. Modified commercially-available smartwatches (Samsung Gear S2, Samsung Gear Fit2) with the SpectroPhon DBM attached were affixed to the left and right wrists of study participants prior to physical exertion testing. Each participant was subjected to moderate physical activity by walking on a treadmill. Data from both smartwatches were obtained simultaneously, collected to smartphone data caches through a Bluetooth wireless interface. Participant weights were monitored using a commercially-available digital balance (Shekel B-200-P). All experiments were conducted indoors under ambient temperature (18°C) and humidity (40–60%). The trial was approved by the Institutional Review Board of Tirat Carmel Mental Health Center (Tirat Carmel, Israel) and registered externally with NIH under study NCT03229109 (http://clinicaltrials.gov).

## Experimental groups

Number of participants: 240, age range: 18–70 (120 men, 120 women). Quartile distributions by age were as follows (minimum, 25% percentile, median, 75% percentile, maximum): Men

(19, 26, 38, 50, 70), women (22, 26.5, 38, 49.75, 69). Mean age of participants (±SD): Men, 39.76±13.59; women, 39.90±13.63.

**Inclusion criteria.**

1. Age: 18 or older, both sexes.

2. Ability and willingness to sign an informed consent document for participation in the study.

**Exclusion criteria.**

1. Presence of known cardiovascular disease.

2. Evidence of any other serious medical disorder.

3. Pregnancy.

## Procedure

Participants were weighed in triplicate prior to, during each rest break, and after the experiment (no clothing after maximal drying). Study volunteers were subjected to 15 minutes of physical activity (walking on the treadmill) with intermittent, timed rest breaks of 10 minutes. Participant skin was examined after the procedure to monitor any allergic reaction or any other skin reaction related to placement of the DBM.

**Activity protocol.** Total time for the experiment was 90 minutes, with a total exercise time of 60 minutes in 15-minute increments. The following exercise and rest intervals were used (times in minutes [min]):

T0: Initiate exercise; T1: T0+15 min—stop exercise, rest; T2: T0+25 min—initiate exercise; T3: T0+40 min–stop exercise, rest; T4: T0+50 min—initiate exercise; T5: T0+65 min–stop exercise, rest; T6: T0+75 min—initiate exercise; T7: T0+90 min–stop exercise. Total duration of study: 90 min.

**Intensity of exercises.** Participants could choose high or low intensity of exertion in each exercise interval based on their level of comfort. We used the following pre-programmed combinations of treadmill speeds (in minutes) for each exercise interval:

a. **High**: 0:00–0:01 –preparation; 0:01–0:05–5.5 km h$^{-1}$; 0:05–0:10–6.0 km h$^{-1}$; 0:10–0:15–6.5 km h$^{-1}$

b. **Low**: 0:00–0:01 –preparation; 0:01–0:05–5.0 km h$^{-1}$; 0:05–0:10–5.5 km h$^{-1}$; 0:10–0:15–6.0 km h$^{-1}$

The objective of the selected treadmill speed regimens was to gradually transition study participants to speed walking without gait transition to running through a series of speed increases. Treadmill speeds were selected based on the difference between an average preferred walking speed of 1.4 m/s [14] and a running gait transition speed of 2.0 m/s [15], divided into 4 even speed increments. The bottom 3 speeds (5.04, 5.58, 6.12 m/s) were designated the "Low" series and the top 3 speeds (5.58, 6.12, 6.66 m/s) were designated the "High" series. The minimal speed increment of the treadmill model used here was 0.5 km/h; we selected treadmill speeds in km/h that approximated our walking speed increment calculations.

## Data recording

The DBM application recorded sweat mass and total salt in sweat every 20 s and automatically transmitted results to a data archive on a Bluetooth-linked mobile phone. Manual recording of

participant weight by use of a digital medical balance (no clothing after maximal drying) was conducted prior to test initiation and during rest breaks (between phases T1-T2, T3-T4, T5-T6, and after T7). During the procedure, participants could drink up to 500 mL of water. The weight of the bottle was measured and recorded after drinking during breaks using a digital laboratory balance (Ohaus V51P6). Mass of water consumed was used to adjust estimated body mass water loss. Participants could not urinate after T0 until the end of trial. For participant safety, we ensured that water weight loss did not exceed 2% of initial measured body mass during the experiment. *Participants could cancel the experiment at any point of the procedure if desired.*

## Statistics

SpectroPhon DBM data output and corrected participant water mass loss were analyzed by Pearson correlation. Data were also used to construct Bland-Altman plots (difference vs. average) for method agreement value distributions as well as frequency distributions (difference) with accompanying skewness and kurtosis estimates (using a D'Agostino-Pearson Omnibus K2 test). The following calculations were performed to compare the DBM and manual weight results for method agreement: mean bias, mean absolute percentage error (MAPE), percent normalized root mean square error (%NRMSE), and mean absolute error (MAE). All statistics were performed using GraphPad Prism 7.0 or Microsoft Excel. Formulae used for calculations are provided in S1 Table. Non-identifying human participant data is available in a public dataset archive [16].

## Results

Most participants (97%) chose high intensity level of exertion. Only 1 participant was not able to finish the procedure due to a prior leg trauma (not related to the current experiment). In the first days of the experiment, there were difficulties with data recording from the SpectroPhon DBM incorporated in the Samsung Gear Fit2 due to conflict between DMB software and software monitoring energy consumption. The problem was quickly solved by a DBM software update. No adverse skin reactions were observed in any participant following the test.

Pearson correlation analyses of method agreement for DBM-estimated water loss (perspiration) compared with the weight change standard used here (participant change in mass) yielded Pearson rho (ρ) values (Fig 2) ranging from 0.8885 (for men wearing the DBM Samsung Gear Fit2; Fig 2E) to 0.9511 (for women wearing the DBM Samsung Gear S2; Fig 2C). All Pearson correlations showed significant positive method correlations (p<0.0001).

Measurement method comparisons by Bland-Altman plots (difference vs. average) indicated normal Gaussian distributions around the line of equality for all participants ([Bias±SD]: 7.531±46.81, DBM Samsung Gear S2, Fig 3A; 3.435±49.73, DBM Samsung Gear Fit2, Fig 4A), as well as when compared by men only ([Bias±SD]: 8.719±53.13, DBM Samsung Gear S2, Fig 3B; 3.261±56.72, Samsung Gear Fit2, Fig 4B) and women only ([Bias±SD]: 6.342±39.70, DBM Samsung Gear S2, Fig 3C; 3.608±41.84, Samsung Gear Fit2, Fig 4C). Differences between measurement methods for all participants were low, with only 4.58% (Gear S2) and 4.17% (Gear Fit2) of DBM estimates falling outside (outlier values) of the 95%CI for the limits of agreement. Outlier method difference values for men were remarkably low at 1.67% for both devices, whereas method difference values for women tended to be higher (4.17%, Gear S2; 3.33%, Gear Fit2).

Frequency distributions of difference values by 30 g bins again yielded normal, Gaussian value distributions for nearly all participants (DBM Samsung Gear S2: K2 = 5.163, p = 0.0756; skewness = -0.2977; kurtosis = -0.3552; Fig 3D; DBM Samsung Gear Fit2: K2 = 0.5942,

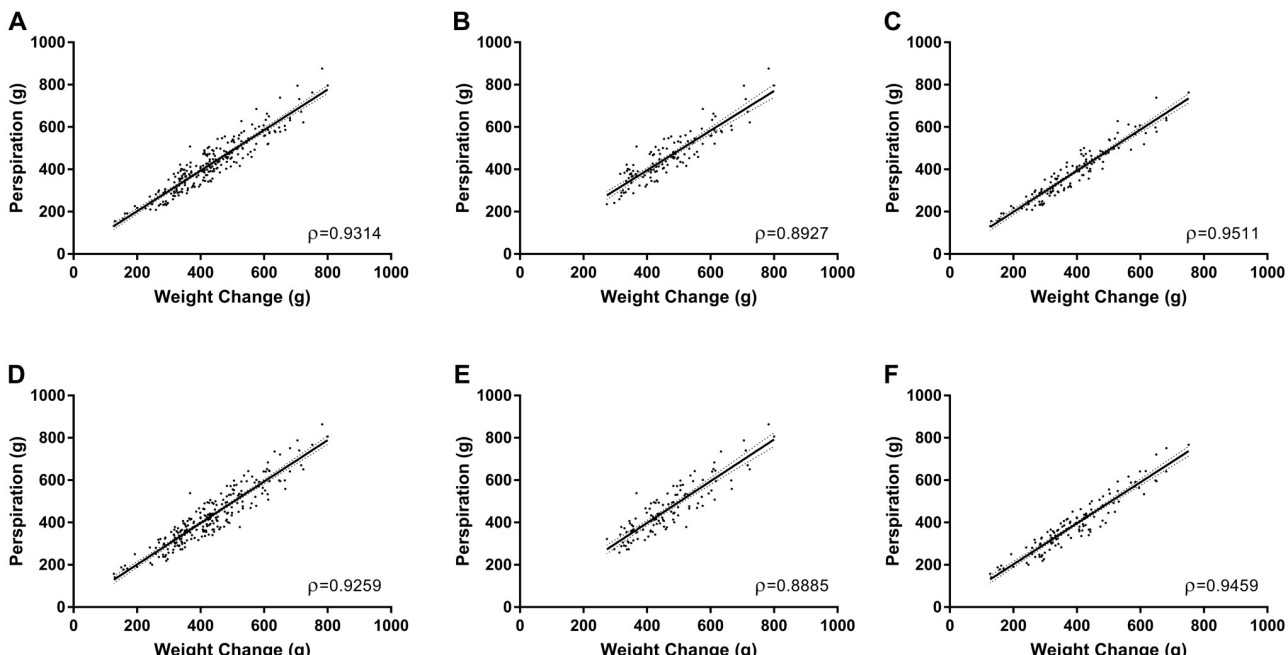

**Fig 2. Comparison of participant weight change with perspiration.** Data represent Pearson (ρ) correlation with accompanying linear regression line of final weight change (g) versus DBM device-measured water loss (perspiration, g). Samsung Gear S2: (A) All participants. (B) Men. (C) Women. Samsung Gear Fit2: (D) All participants. (E) Men. (F) Women. Solid line, regression best fit; dotted lines, upper and lower bounds of 95%CI of the regression line. Correlation, all measures: p<0.0001. For all regressions, there were no significant deviations from linearity (slopes were all significantly non-zero, p<0.0001).

p = 0.7430; skewness = -0.0401; kurtosis = -0.2344; Fig 4D), as well as when compared by men only (DBM Samsung Gear S2: $K2 = 4.980$, $p = 0.0829$; skewness = -0.3606; kurtosis = -0.5397; Fig 3E; DBM Samsung Gear Fit2: $K2 = 1.3310$, $p = 0.5140$; skewness = -0.1826; kurtosis = -0.3428; Fig 4E) and women only (DBM Samsung Gear S2: $K2 = 2.2920$, $p = 0.3179$; skewness = -0.1967; kurtosis = -0.4630; Fig 3F). The only non-Gaussian exception was the difference distribution of method agreement for women wearing the DBM Samsung Gear Fit2 ($K2 = 10.0100$, $p = 0.0067$; skewness = 0.3288; kurtosis = -0.7943; Fig 4F), which presented a narrowed, peaked distribution that was slightly left-skewed.

Table 1 summarizes method comparison statistics for the DBM Samsung Gear S2 device and the standard (mass loss) measurement method. Mean bias percentage for all participants was low (1.77%) and similar values were measured for men (1.87%) and women (1.63%), indicating close method agreement. MAPE values were similarly low, approximately 10%, also indicating that the DBM device output to the Samsung Gear S2 smartwatch was also in close agreement with our standard mass loss measurement method (mean±95%CI: all participants, 9.56±0.91; men, 10.16±1.36; women, 8.96±1.23). The %NRMSE estimation of method difference was also low for all participants (2.11%) with similar values for both men (1.87%) and women (2.50%). The MAE estimates between methods were as follows: [mean(g) ±SE]; all participants, 39.51±1.68; men, 45.81±2.55; women, 33.22±2.05.

Table 2 summarizes method comparison statistics for the DBM Samsung Gear Fit2 device and the standard (mass loss) measurement method. Mean bias percentage for all participants was lower than seen with the DBM Samsung Gear S2 device (0.80%) and similar values were measured for men (0.70%) and women (0.93%), indicating close method agreement. MAPE values were similarly low, again approximately 10%, also indicating that the DBM device

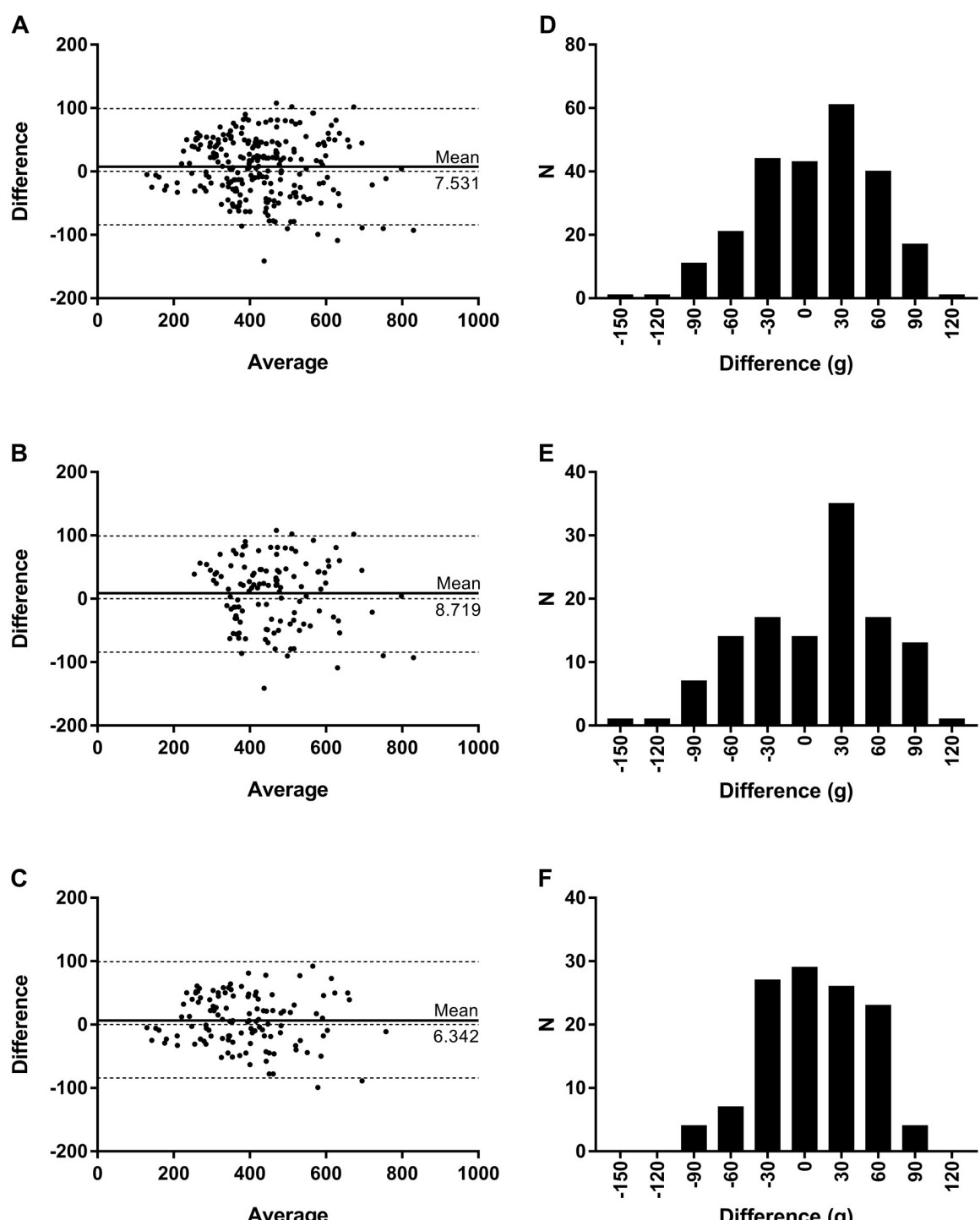

**Fig 3. Device-standard agreement for Samsung Gear S2.** (A-C) Bland-Altman plots of average versus difference for perspiration measurements of DBM Samsung Gear S2 (g) compared against participant weight change (g) for (A) all participants, (B) men, and (C) women. Solid line, line of equality; dotted lines, upper and lower bounds of 95%CI of the line of equality. (D-E) Frequency distribution histograms of method measurement differences (g) for (D) all participants, (E) men, and (F) women.

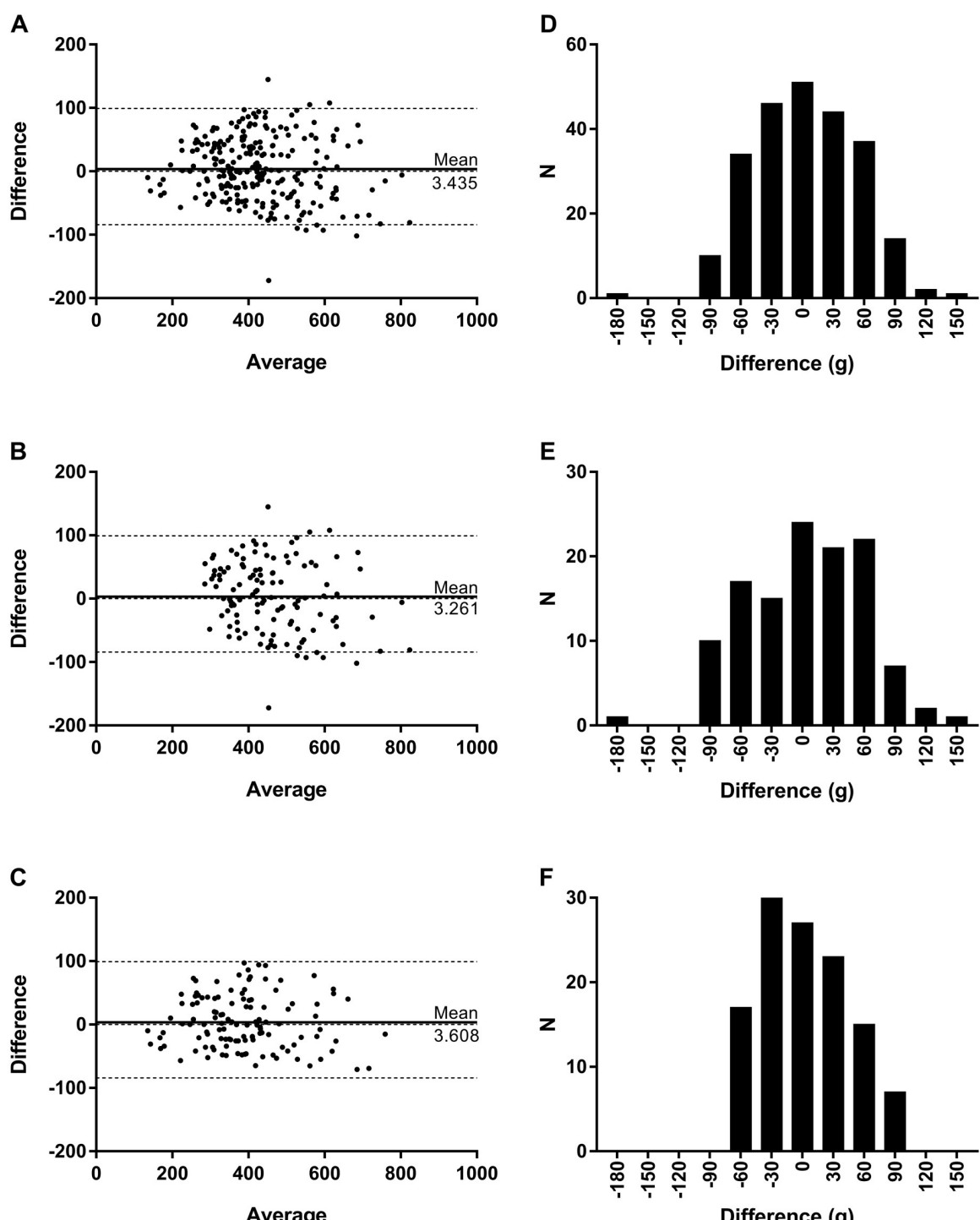

**Fig 4. Device-standard agreement for Samsung Gear Fit2.** (A-C) Bland-Altman plots of average versus difference for perspiration measurements of DBM Samsung Gear Fit2 (g) compared against participant weight change (g) for (A) all participants, (B) men, and (C) women. Solid line, line of equality; dotted lines, upper and lower bounds of 95%CI of the line of equality. (D-E) Frequency distribution histograms of method measurement differences (g) for (D) all participants, (E) men, and (F) women.

**Table 1. Summary of statistical comparisons between DBM Samsung Gear S2 and the mass loss measurement standard method.**

| | | All Participants | Men | Women |
|---|---|---|---|---|
| **Mean Bias** | g | 7.53 | 8.72 | 6.34 |
| | % | 1.77 | 1.87 | 1.63 |
| **MAPE** | Mean | 9.56 | 10.16 | 8.96 |
| | 95%CI | 0.91 | 1.36 | 1.23 |
| | %SD | 7.58 | 7.58 | 6.86 |
| **%NRMSE** | | 2.11 | 1.87 | 2.50 |
| **MAE** | g | 39.51 | 45.81 | 33.22 |
| | SE | 1.68 | 2.55 | 2.05 |

Abbreviations are as follows: MAPE, mean absolute percentage error; %NRMSE, percent normalized root mean square error; MAE, mean absolute error; 95%CI. 95% confidence interval; %SD, percent standard deviation; SE, standard error of the mean.

output to the Samsung Gear Fit2 smartwatch was in close agreement with our standard measurement method (mean±95%CI: all participants, 9.92±0.96; men, 10.32±1.38; women, 9.52 ±1.34). The %NRMSE estimation of method difference was also low for all participants and similar to the DBM Samsung Gear S2 measures (2.01%) with similar values for both men (1.77%) and women (2.39%). The MAE estimates between methods were as follows: [mean(g) ±SE]; all participants, 40.94±1.83; men, 47.11±2.87; women, 34.78±2.13.

There were no significant differences in MAPE or MAE values for all participants, men, or women between the two SpectroPhon DBM devices (Student's t-test, 2-tailed unpaired: $p>0.05$ for all comparisons).

## Discussion

In mammals, estimation of normal body hydration (euhydration) is approximated by the ratio of total body water mass to fat-free tissue mass, which is 0.73 in nearly all cases [17, 18]. Both terrestrial and pelagic species demonstrate this same ratio, thus indicating that body water maintenance is governed by mechanisms consistent across mammalian taxa and is therefore of central importance to basic metabolism and excretory processes. Typical homeostatic limits of total body water content during rest are within 0.22% of body mass but can vary as much as 0.48% of body mass under conditions of rigorous physical exertion and heat stress. For

**Table 2. Summary of statistical comparisons between DBM Samsung Gear Fit2 and the mass loss measurement standard method.**

| | | All Participants | Men | Women |
|---|---|---|---|---|
| **Mean Bias** | g | 3.43 | 3.26 | 3.61 |
| | % | 0.80 | 0.70 | 0.93 |
| **MAPE** | Mean | 9.92 | 10.32 | 9.52 |
| | 95%CI | 0.96 | 1.38 | 1.34 |
| | %SD | 7.58 | 7.72 | 7.47 |
| **%NRMSE** | | 2.01 | 1.77 | 2.39 |
| **MAE** | g | 40.94 | 47.11 | 34.78 |
| | SE | 1.83 | 2.87 | 2.13 |

Abbreviations are as follows: MAPE, mean absolute percentage error; %NRMSE, percent normalized root mean square error; MAE, mean absolute error; 95%CI. 95% confidence interval; %SD, percent standard deviation; SE, standard error of the mean.

humans, the weekly average total body water mass variation is approximately 2% of body mass based primarily on changes in hydration level and body fat content [4]. When total body water content drops below 10% body mass, several compensatory mechanisms engage to drive water- and salt-seeking behaviors, which is known as thirst response [4, 19, 20].

The thirst response is complex and often involves a variety of psychological and social cues in addition to physiological thirst stimulus [4, 21]. Changes in blood osmolality stimulate osmoreceptors in hypothalamus, increasing release of antidiuretic hormone, and also increase secretory responses to reduced blood flow in kidney (increased renin) and lung (increased angiotensin converting enzyme). These compensatory physiological mechanisms can impose restrictions on glomerular function and foster water and sodium retention, yet they only have partial influence on water-seeking behavior [21].

The just noticeable difference threshold for human thirst perception has been estimated to be at about 1–2% from optimal hydration [4], a rather low threshold, however procrastination in addressing water hunger is frequently observed and mitigated by situational and social elements [21]. These complicating elements, which distract from thirst response perception, include water availability, fluid source taste, developed drinking habits, and association with meals [5]. Therefore, in many cases the thirst response is usually perceived when the stimulus becomes strong enough to override other environmental distractions and becomes more of an indicator of definitive immediate need [7]. Physiological evidence from athlete hydration levels during and after rigorous exercise, their perceived thirst level, and drive to drink water indicates that thirst is an approximation stimulus of hydration condition only and may not adequately lead hydration state to provide properly-timed behavioral compensation [22, 23]. Indeed, engaging in rigorous exercise in a hypohydrated state and imbibing water afterward during rest suppresses the thirst stimulus and drive to consume additional fluids, despite a continued state of hypohydration [7]. As such, perceived thirst cannot be reliably used as an indicator of hydration state until critically low hydration levels are reached [7, 8].

Typical methods of measuring hydration involve some form of hematological or other body fluid assessment. There are many methods with the more commonly-used measures being hematocrit, plasma, saliva, or urine osmolality, sodium and potassium concentration of sweat, and level of blood gas carbonates [24, 25]. All of these methods, however, require either laboratory processing or some form of biosensor to measure constituents of collected fluid in real time. The disadvantage of fluid collection-based approaches is the necessity to collect and store fluid, even if temporarily. This typically requires either absorptive pads or some form of bulky microfluidic device, both of which have a limited span of use. In contrast, methods of measuring skin perspiration that are amenable to "wearables" fall into either of two general classes of device: electrode-based devices that contact with the biofluid or polymer-based sensors that react to the presence of specific constituents of sweat [26]. Surface-reactive films, whether optical, chemical, or electrode-based, avoid the need for fluid collection, have extended use potential, tend to be less bulky, and are more portable and less energy-consumptive for use in real-time data capture devices [27]. For an extensive review of wearable device technologies and their applicable chemosensory use, despite being laboratory demonstration devices, see Yang and Gao, 2019 [28].

In the present study, we examined the accuracy of two SpectroPhon DBM devices in a group of human volunteers engaged in moderate physical exercise. The DBM is a polymer film-based photoplethysmographic (PPG) device that measures sodium ion concentration in sweat and galvanically estimates whole body skin area to provide an estimation of total body water loss in real time. Synchronized to a smartphone with a data interpretation application, the pairing allows for continuous monitoring and post exercise analysis. Performance comparisons of the DBM with similar commercially-available devices were not possible here since,

despite the great interest in wearable hydration monitors, only one other commercially-available product exists for which there is no published data (the Kenzen). The majority of hydration sensor studies cover laboratory calibration efforts only and there exist no published wearable hydration monitor field tests.

Among the measured method agreement metrics for the SpectroPhon DBM-modified smartwatches examined here, the method error for all groups studied ranged from 2.01–2.50%, far below the acceptable measurement method error (15% cutoff by the ISO15 standard for glucometers) of other SpectroPhon devices we have examined previously [29]. The low error values calculated here (around 2% for %NRMSE; mean bias <2%) quite accurate comparing the different methods of measurement. Confirming this conclusion is the finding that <5% of differences between measurement methods for all subjects by Bland-Altman analyses fell outside of the 95%CI for the limits of agreement. When the results are considered collectively, we feel that the PPG technology examined here has excellent potential as a reliable wearable hydration monitor.

## Supporting information

**S1 Table. Measurement method comparison formulae.** Calculated metrics were as follows: Mean bias including mean bias %, MAPE (mean absolute percentage error) including MAPE standard deviation (MAPE SD) and 95% confidence interval (MAPE 95%CI), %NRMSE (percent normalized root mean square error), MAE (mean absolute error) including MAE standard deviation (MAE SD). Formula abbreviations are as follows: W = Smartwatch-DBM measurement; B = balance standard comparison measurement.
(PDF)

**S1 Checklist. TREND statement checklist.**
(PDF)

**S1 File.**
(PDF)

**S2 File.**
(PDF)

**S3 File.**
(PDF)

**S4 File.**
(PDF)

**S5 File.**
(PDF)

## Acknowledgments

We would like to thank Dr. Natali Sedugin (Maale HaCarmel Mental Health Center, affiliated to the Rappaport Faculty of medicine, Technion, Haifa, Israel) for their assistance in conducting this study.

## Author Contributions

**Conceptualization:** Dmitry Rodin, Yair Shapiro, Albert Pinhasov, Anatoly Kreinin.

**Data curation:** Dmitry Rodin.

**Formal analysis:** Michael Kirby.

**Investigation:** Dmitry Rodin, Michael Kirby.

**Project administration:** Anatoly Kreinin.

**Supervision:** Dmitry Rodin, Yair Shapiro, Albert Pinhasov, Anatoly Kreinin.

**Writing – original draft:** Michael Kirby.

**Writing – review & editing:** Michael Kirby.

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
