## [Decision Letter · Decision Letter 0]

3 May 2022

PONE-D-22-00744An accurate wearable hydration sensor: Real-world evaluation of practical usePLOS ONE

Dear Dr. Kirby,

Thank you for submitting your manuscript to PLOS ONE. I sincerely apologize for the delays in the peer review of the manuscript. After careful consideration, we feel that it has merit but does not fully meet PLOS ONE’s publication criteria as it currently stands. Therefore, we invite you to submit a revised version of the manuscript that addresses the points raised during the review process.

Please review and respond to each reviewer comment. I look forward to your revised submission.

We look forward to receiving your revised manuscript.

Kind regards,

Matthew M. Schubert, Ph.D.

Academic Editor

PLOS ONE

Journal Requirements:

There was no funding provided for this project. "The funders had no role in study design, data collection and analysis, decision to publish, or preparation of the manuscript"

Additional Editor Comments:

Please review and respond to the critiques raised by each reviewer.

Reviewers' comments:

Reviewer's Responses to Questions

**Comments to the Author**

1. Is the manuscript technically sound, and do the data support the conclusions?

Reviewer #1: Yes

Reviewer #2: Partly

2. Has the statistical analysis been performed appropriately and rigorously? 

Reviewer #1: Yes

Reviewer #2: Yes

3. Have the authors made all data underlying the findings in their manuscript fully available?

Reviewer #1: Yes

Reviewer #2: Yes

4. Is the manuscript presented in an intelligible fashion and written in standard English?

Reviewer #1: Yes

Reviewer #2: Yes

5. Review Comments to the Author

Reviewer #1: The main claims of the paper are significant for the discipline and are properly placed in the context of the previous literature

Data and analyses fully support the claims.

The manuscript is well organized and written clearly.

The work is original and of great scientific relevance.

Up-to-date and relevant bibliography.

The study presents and confirms a reliable tool to assist in the monitoring of hydration in the geriatric population and in a physical training scenario. Other populations may also benefit from this tool.

Reviewer #2: Many thanks for the opportunity of reviewing your manuscript.

The study investigated the accuracy of a novel hydration sensor combining photoplethysmographic and galvanic technologies to monitor hydration status in 240 subjects from a wide age range. In addition, participants' fluid mass loss was assessed by determining sweat rate and using the body hydration sensors attached to commercially available smartwatches.

Assessing hydration status can be very challenging. Therefore, combining methods (such as changes in body mass, urine markers, blood markers, etc.) is always recommended to obtain valid and reliable results. While the concept of this device is very innovative, the study has some methodological and physiological aspects of being considered:

- It would be essential to include the participants' fitness levels and body composition (if data are available) to understand their physiological responses better.

- The exercise regimes do not look too different from each other. It would be important to explain why those treadmill speed combinations were selected. What is the rationale for the rest breaks? What would be the effect of these pauses in thermoregulation responses?

- Several studies have observed differences in sweat composition and sweat rates in different regions of the body (Baker LB et al., 2016; Barnes KA et al., 2019; Rivera Brown AM, 2020; Baker LB 2017); what is the rationale for wearing the sensor in the wrist? How does it correlate with whole-body sweat?

- What would be the differences or advantages of the SP-DBM over the microfluidic devices? (Baker LB et al., 2020).

- Was the determination of sodium concentration in sweat through the SP-DBM validated using methods such as flame photometry?

One of the main strengths of this paper is that it proposes a new method to assess hydration in the population. Figure 1 is very clear and self-explanatory. However, some of the weaknesses are the lack of clarity in some paragraphs of the methods and analysis. It also would be interesting to include some reliability data if available. It also would be essential to include a hypothesis as part of the introduction. The introduction could also include a more extensive literature review of the most recent devices and sensors to assess hydration status in the athletic population. It is also not clear why the authors referenced older adults admitted to hospitals and concluded in their abstract that the device could be reliable (?) in the geriatric population when most of the participants did not belong to this demographic group. Some of the details regarding the participants' laboratory visits should be expanded and clarified to avoid confusion regarding what the researchers investigated. The start of the discussion can be restructured, since there is much information regarding thirst (which is interesting); however, it is not relevant to the results of this study. The discussion could also include a critique of the advantages and disadvantages of the SP-DBM over other methods and compare results. No limitations of the study were mentioned in the discussion sections. Some relevant references (like those mentioned earlier) are missing and should be included in the manuscript. Some parts of the manuscript can be written in a more precise and formal manner to improve readability.

6. PLOS authors have the option to publish the peer review history of their article (what does this mean?). If published, this will include your full peer review and any attached files.

Reviewer #1: **Yes: **Sandra Celina Fernandes Fonseca

Reviewer #2: No

---

## [Author Response · Author response to Decision Letter 0]

15 Jun 2022

Matthew , Schubert, PhD

Academic Editor

PLoS One

14 June 2022

Michael Kirby, PhD

Department of Molecular Biology and Adelson School of Medicine

Ariel University

Israel

Dear Dr. Schubert,

As requested, we provide here a point-by-point response to reviewer comments. We thank the reviewers for their thorough reading of our research article. Below are specific changes we have made to address their concerns and feel that the quality of the manuscript has been substantially improved.

Best regards,

Michael Kirby

We have renamed the files to meet with PLoS One conventions.

There was no funding provided for this project. "The funders had no role in study design, data collection and analysis, decision to publish, or preparation of the manuscript"

The authors received no specific funding for this work.

We have added the following references to the manuscript:

14. Browning RC, Baker EA, Herron JA, Kram R. Effects of obesity and sex on the energetic cost and preferred speed of walking. Journal of applied physiology. 2006;100(2):390-8. doi: 10.1152/japplphysiol.00767.2005. PubMed PMID: 16210434.

15. Mohler BJ, Thompson WB, Creem-Regehr SH, Pick HL, Jr., Warren WH, Jr. Visual flow influences gait transition speed and preferred walking speed. Experimental brain research. 2007;181(2):221-8. doi: 10.1007/s00221-007-0917-0. PubMed PMID: 17372727.

Comments to the Author

1. Is the manuscript technically sound, and do the data support the conclusions?

Reviewer #1: Yes

Reviewer #2: Partly

We do not fully understand the justification of Reviewer #2 stating that the manuscript is “partly” sound or that the data “partly” support the conclusions. We assume the criticism lies with some methodological omissions, which have been remedied (please see below).

2. Has the statistical analysis been performed appropriately and rigorously?

Reviewer #1: Yes

Reviewer #2: Yes

3. Have the authors made all data underlying the findings in their manuscript fully available?

Reviewer #1: Yes

Reviewer #2: Yes

4. Is the manuscript presented in an intelligible fashion and written in standard English?

Reviewer #1: Yes

Reviewer #2: Yes

5. Review Comments to the Author

Reviewer #1: The main claims of the paper are significant for the discipline and are properly placed in the context of the previous literature

Data and analyses fully support the claims.

The manuscript is well organized and written clearly.

The work is original and of great scientific relevance.

Up-to-date and relevant bibliography.

The study presents and confirms a reliable tool to assist in the monitoring of hydration in the geriatric population and in a physical training scenario. Other populations may also benefit from this tool.

We thank Sandra Celina Fernandes Fonseca, Reviewer #1, for her time and considerate review of our work.

Reviewer #2: Many thanks for the opportunity of reviewing your manuscript.

The study investigated the accuracy of a novel hydration sensor combining photoplethysmographic and galvanic technologies to monitor hydration status in 240 subjects from a wide age range. In addition, participants' fluid mass loss was assessed by determining sweat rate and using the body hydration sensors attached to commercially available smartwatches.

Assessing hydration status can be very challenging. Therefore, combining methods (such as changes in body mass, urine markers, blood markers, etc.) is always recommended to obtain valid and reliable results. While the concept of this device is very innovative, the study has some methodological and physiological aspects of being considered:

We thank Reviewer #2 for their time and consideration of this work, as well as their specific criticisms regarding the reporting of our study methodology. We have attempted to amend the manuscript to address these concerns and provide greater clarity.

Prior to rendering any rebuttal statements below, we feel that there is some confusion on the part of the reviewer regarding our relationship to SpectroPhon. We are academic scientists and physicians and are not affiliated with the SpectroPhon corporation. Furthermore, we were not involved in the device design or method validation in their laboratories. Therefore, much of the requested information by the reviewer is not available to us as it constitutes SpectroPhon proprietary data. The diagrams of the SP-DBM provided to the reader in Figure 1 were gleaned from publicly-available SpectroPhon patent filings.

Regarding hydration status of study volunteers, this was not a study objective. This report constitutes a field trial evaluating device performance across a range of volunteer users with varying BMI, hydration state, fitness level, and age. This was an attempt to examine the device accuracy compared against a simple, standard method of estimating body mass loss to determine whether the device was applicable for broad consumer use in the general population.

(1) It would be essential to include the participants' fitness levels and body composition (if data are available) to understand their physiological responses better.

Participant sex, age, and weight are available and provided to the reader in the data archive. Those data may be calculated by the reader; however, we did not feel that BMI and fitness level was germane to the intent of the study.

The objective of the study was not to assess differential physiological responses based on body morphology, cardiovascular fitness, activity level, hydration level, blood sugar regulation, or any other parameter apart from changes in body mass following moderate exertion. All participants met the basic criteria for inclusion in the study: Adults who could provide written consent and were not pregnant or had any known cardiovascular disease or other serious health condition.

(2) The exercise regimes do not look too different from each other. It would be important to explain why those treadmill speed combinations were selected. What is the rationale for the rest breaks? What would be the effect of these pauses in thermoregulation responses?

We thank the reviewer for this suggestion. Preferred walking speed, even in obese subjects, is approximately 1.4 m/s (after Browning et al [2006] J Appl Physiol 100:390) with gait transition speeds to running at approximately 2.0 m/s (after Mohler et al [2007] Exp Brain Res 181:221). We attempted to divide the difference between these two speeds into 4 even increments with the bottom 3 rated as “low” and top 3 rated as “high” to provide options for participant comfort. Our objective was to transition participants into speed walking but remain below the threshold for gait transition. The speeds we selected were based upon the minimum increment speed adjustment of the treadmill used for the experiment (0.5 km/h), which approximated the m/s speed increments we calculated. Browning’s group used 5-minute rest breaks; we elected to double this time since the age range of subjects was broad, we were asking them to exercise for 90 minutes, and sufficient time was required for participants to disrobe, be weighed, drink water, and redress to continue exercise.

We added in a brief justification of treadmill speeds with references to explain to the reader our design of the exercise regimens.

Thermoregulatory responses during rest breaks would presumably allow for evaporative cooling and for subjects to take water and feel refreshed enough to continue with the test. This was our intent. Please bear in mind that the human participants were not the subject of this study. The two test devices were the subjects. Therefore, we were simply interested in how the device performance correlated with a reference method of estimated body mass loss.

(3) Several studies have observed differences in sweat composition and sweat rates in different regions of the body (Baker LB et al., 2016; Barnes KA et al., 2019; Rivera Brown AM, 2020; Baker LB 2017); what is the rationale for wearing the sensor in the wrist? How does it correlate with whole-body sweat?

We imagine the primary reason for wearing the sensors on the wrist is a simple matter of utility. Regarding the rationale for wearing the sensor on the wrist as opposed to other body locations, we tested the two devices in the manner in which they are intended to be used. Please keep in mind that we are not employees of SpectroPhon, did not manufacture the modified smartwatches, and did not have permission to modify the prototypes in any way. In short, they were not our property.

If whole body sweat and evaporative loss corresponds with body mass loss, then the device outputs were accurate in predicting those changes. Since participants during the test could not urinate or defecate, were weighed before, during, and after in the nude, and any water consumed was weighed and adjusted to the medical scale measurements, one can safely assume that the body mass loss can be attributed to water loss through perspiration and respiration.

With respect to differences in sweat composition, SpectroPhon has been testing sweat kinetics for many years and based on prior studies (the results of which have been published; see Zilberstein et al [2018] Electrophoresis 39:2344 and Rodin et al [2019] Clin Biochem 65:15), they concluded that the wrist is the optimal location for measurements. Different parts of the body do have different sweat kinetics and sweat composition differs between individuals in regard to organic compounds. However, the major fraction content (water and NaCl) is very stable and predictable between individuals. The sensors also measure other sweat components which are included in their translation algorithms, however SpectroPhon would not share that information with us.

(4) What would be the differences or advantages of the SP-DBM over the microfluidic devices? (Baker LB et al., 2020).

The advantages of microfluidic devices are that whole sweat can be collected and various sweat analytes can be assayed. This is useful for analytical physiology studies and is more amenable to laboratory use. Most microfluidic devices have a limited use due to restrictions in chamber volume and limited lifespan in that there are challenges in cleaning and repurposing those devices. The SP-DBM does measure a range of sweat analytes and could potentially be retasked to be used in much the same manner, however in their present form they are intended as consumer devices with perhaps some future medical uses as well. The chemochromic film of the SP-DBM, which has the appearance of a kaleidoscopic sticker or adhesive sheet, also has a limited lifespan due to chemical reactions in the crystalline structure (SpectroPhon has stated they estimate a use life of 4-6 months before the film would need replacement). The main advantages of the SP-DBM are their use lifespan, utility as a wearable, and their accuracy.

(5) Was the determination of sodium concentration in sweat through the SP-DBM validated using methods such as flame photometry?

We understand that this was performed for sodium and other ions in sweat in the SpectroPhon laboratories. These data were used to construct some complex algorithms to incorporate several sweat components to estimate water volume on the sensor.

(6) One of the main strengths of this paper is that it proposes a new method to assess hydration in the population. Figure 1 is very clear and self-explanatory. However, some of the weaknesses are the lack of clarity in some paragraphs of the methods and analysis.

Without specific examples, we cannot fully address these concerns. We feel the methods are rather straightforward and all analytical formulae used are detailed in Table S1, however we have made several additions for enhanced clarity. Methods regarding our choice of treadmill speeds has been added. Further, we specify that participants were weighed in triplicate in the nude prior to start, during each rest break, and at exercise conclusion.

(7) It also would be interesting to include some reliability data if available.

We agree. However, as we have stated, those data are not publicly available and we are not affiliated with SpectroPhon.

(8) It also would be essential to include a hypothesis as part of the introduction.

Here we state the two objectives of the study. “The main objective of current study is to estimate the accuracy of SpectroPhon perspiration biosensors incorporated in two smartwatches: a Samsung Gear S2 and a Samsung Gear Fit2. The secondary aim of the study is to also evaluate the safety-in-use of SpectroPhon biosensors.”

This type of study is not particularly hypothesis-driven as we did not know what to expect. We could neither speculate that the SP-DBM would accurately predict body water loss or not predict loss. To state that we “predict that the SP-DBM will closely mirror body water loss assessed by a standard, scale-based method” would be inaccurate, as we had no prior data upon which to base that hypothesis. We did not make any assumptions regarding the device performance.

(9) The introduction could also include a more extensive literature review of the most recent devices and sensors to assess hydration status in the athletic population.

The purpose of the introduction was to lead the reader into understanding that a need exists for convenient sweat monitoring devices. We elected to mention other devices and methodology briefly in the introduction and discussion sections, only providing review citations. We did not delve into an extensive discussion of alternative methodologies for several reasons.

(1) To do justice to the array of devices used for hydration monitoring and/or sweat constituent analysis would essentially double the size of the manuscript and leave the reader confused as to whether this is a research article or a review. For example, we would need to discuss and cite numerous works by the following research groups (listed by PI): W Gao; J Rogers; A Abbaspourrad; D-H Kim; H Xu and Z Gu; A Javey; Y-H Cho; T Kaya; C-H Ting; B-R Li; Z Zhao; ASM Steijlin; W Cheng; T Sakata; C-M Chen, S Anastasova, K Zhang, BG Rosa, BPL Lo, HE Assender, and G-Z Yang; C Zhao and H Liu; S Emaminejad; A Karyakin; F Andrade; H Alshareef; JY Park; O Parlak; T Pan. As the article stands now, the body text is 3500 words. We feel for the amount of data presented, that article length is sufficient.

(2) There are no other commercial (as in purchasable) devices for hydration monitoring other than an industrial wearable by Kenzen (for which there is no data or publication). We performed an extensive literature search and all other sweat monitoring devices as of October 2021 (when we submitted this article) are experimental laboratory tools that may never make it out of the laboratory. We did not spend time discussing speculative technologies. Since we are not in the business of device development and were simply evaluating a prototype soon to be commercially released, we did not see a need to engage in extensive review of various methodologies for sweat monitoring.

(10) It is also not clear why the authors referenced older adults admitted to hospitals and concluded in their abstract that the device could be reliable (?) in the geriatric population when most of the participants did not belong to this demographic group.

If you examine the age distribution of participants, we attempted to assemble a balanced population with equal numbers of each sex and nearly equal numbers of participants by age category. We did not aim to examine the use of these devices in a geriatric population, per se, but a representative cross-section of the adult populaiton. Age subcategory analysis did not reveal any trends, indicating that there was no age bias in sensor output. The reference to use in the geriatric population (defined in Israel as 61.5 or older, for which we had 21 study volunteers) was merely speculative. If the device is accurate in predicting body water loss irrespective of age, it follows that it could have a use in hospice or outpatient populations. However, at the present time we understand that it will be marketed as a consumer product and not a medical device.

(11) Some of the details regarding the participants' laboratory visits should be expanded and clarified to avoid confusion regarding what the researchers investigated.

Without sufficient details, we cannot address this point. There were no laboratory visits. All tests were conducted over a 3-month period in a gymnasium using the same treadmill. Recruitment was by public advertising, self-selection, and walk-in. As stated above, we made some detailed additions to the methods section. We hope these additions are sufficient in providing the reader a clearer description of the study design and participant monitoring methods.

(12) The start of the discussion can be restructured, since there is much information regarding thirst (which is interesting); however, it is not relevant to the results of this study.

Respectfully, we disagree. The intent of reviewing the thirst response was to dispel the common misunderstanding that thirst is a lock-step indicator of hydration level. It is an approximation stimulus and can be suppressed by other physiological and psychosocial stimuli. The end of this argument is that for many applications, personal fitness training, athletic training, a reliable and convenient means of estimating hydration level would have great utility over a reliance on thirst response as a hydration indicator.

(13) The discussion could also include a critique of the advantages and disadvantages of the SP-DBM over other methods and compare results.

Since there are no other commercially-available devices with publicly-available data (i.e., Kenzen) apart from experimental laboratory devices, we found any method comparisons difficult. The majority of papers we sourced were engineering manuscripts that primarily focused on method validation. In short, they were reporting their device calibrations and only a few papers provided any data with actual human subjects. The data reported could not be reconciled with our data and were typically in the form of ion concentrations and current output or voltage alterations. As we are not the designers of the SP-DBM, we did not have access to the raw data accumulated by the device to make such comparisons.

(14) No limitations of the study were mentioned in the discussion sections. Some relevant references (like those mentioned earlier) are missing and should be included in the manuscript. Some parts of the manuscript can be written in a more precise and formal manner to improve readability.

We agree that no study limitations are mentioned in the manuscript. However, we must admit that due to the simplicity of the study design with each participant serving as their own control, we could not find any study limitations to report. In the instance where one participant could not complete the full, 90-minute exercise regimen, we simply analyzed the data to the time of discontinuance that we accumulated. There were basically no participant drop-outs by those criteria. We do understand the concept of study limitations well, as we typically publish articles in biological psychiatry and molecular biology and this study is essentially a “one-off” for our group. Nonetheless, we could not determine any foreseeable study limitations.

Regarding the statement that the manuscript be rewritten in a more precise and formal manner, without citing specific examples, we do not have a way to address this critique.

6. PLOS authors have the option to publish the peer review history of their article (what does this mean?). If published, this will include your full peer review and any attached files.

Do you want your identity to be public for this peer review? For information about this choice, including consent withdrawal, please see our Privacy Policy.

Reviewer #1: Yes: Sandra Celina Fernandes Fonseca

Reviewer #2: No

---

## [Decision Letter · Decision Letter 1]

25 Jul 2022

An accurate wearable hydration sensor: Real-world evaluation of practical use

PONE-D-22-00744R1

Dear Dr. Kirby,

We’re pleased to inform you that your manuscript has been judged scientifically suitable for publication and will be formally accepted for publication once it meets all outstanding technical requirements.

Kind regards,

Matthew M. Schubert, Ph.D.

Academic Editor

PLOS ONE

Additional Editor Comments (optional):

Reviewers' comments:

Reviewer's Responses to Questions

**Comments to the Author**

1. If the authors have adequately addressed your comments raised in a previous round of review and you feel that this manuscript is now acceptable for publication, you may indicate that here to bypass the “Comments to the Author” section, enter your conflict of interest statement in the “Confidential to Editor” section, and submit your "Accept" recommendation.

Reviewer #1: (No Response)

Reviewer #2: All comments have been addressed

2. Is the manuscript technically sound, and do the data support the conclusions?

Reviewer #1: Yes

Reviewer #2: Yes

3. Has the statistical analysis been performed appropriately and rigorously? 

Reviewer #1: Yes

Reviewer #2: Yes

4. Have the authors made all data underlying the findings in their manuscript fully available?

Reviewer #1: Yes

Reviewer #2: Yes

5. Is the manuscript presented in an intelligible fashion and written in standard English?

Reviewer #1: Yes

Reviewer #2: Yes

6. Review Comments to the Author

Reviewer #1: (No Response)

Reviewer #2: Thanks for taking the tame to address all my comments and suggestions in a comprehensive and adequate manner. Some of the information that has been provided have allowed me to understand better the purpose and methodology of the study and I think the additions that were made will also help the readers to have a better comprehension of your research.

7. PLOS authors have the option to publish the peer review history of their article (what does this mean?). If published, this will include your full peer review and any attached files.

Reviewer #1: No

Reviewer #2: No

---

## [Editor Report · Acceptance letter]

27 Jul 2022

PONE-D-22-00744R1 

An accurate wearable hydration sensor: Real-world evaluation of practical use 

Dear Dr. Kirby:

I'm pleased to inform you that your manuscript has been deemed suitable for publication in PLOS ONE. Congratulations! Your manuscript is now with our production department. 

Kind regards, 

on behalf of

Dr. Matthew M. Schubert 

Academic Editor

PLOS ONE